

# Response of soil nematode community structure, diversity, and ecological network to elevation gradients in wild fruit forest of Tianshan Mountain, China

Yulu Zhang[1,2,3], Mengyu Yang[1], Wenxin Liu[2,3], Zhicheng Jiang[2,3], Yang Zhao[2,3], Gaofeng Li[2,3], Jing Cao[2,3], Minru Zhang[2,3], Haijun Yang[1,2,3] and Dong Cui[2,3]

[1] Yunnan University, College of Ecology and Environmental Science, Kunming City, Yunnan Province, China

[2] Yili Normal University, Institute of Resources and Ecology, Yining City, Yili Kazak Autonomous Prefecture, China

[3] Yili Normal University, College of Resources and Environment, Yining City, Yili Kazak Autonomous Prefecture, China

Corresponding authors
Haijun Yang, yanghaijun@ynu.edu.cn
Dong Cui, cuidongw@126.com

## ABSTRACT

**Background**. The Tianshan wild fruit forest is a special deciduous broad-leaved forest located in the mountains of central Asia.

**Methods**. To reveal how the Tianshan wild fruit forest ecosystem responds to environmental changes along an elevation gradient, we used the soil nematode index, which is widely recognized as a key indicator of soil health. This study focused on the nematode communities beneath *Juglans cathayensis* and *Malus sieversii*, two typical constructive species in the Tianshan wild fruit forest. Six elevation levels (1,480, 1,401, 1,351, 1,305, 1,252, and 1,207 m) were selected, and a nematode co-occurrence network was constructed for each elevation. We analyzed the abundance, diversity, ecological indices, and network complexity of soil nematode communities across different elevation gradients. In addition, we explored the relationships among environmental factors, soil multifunctionality, and nematode community characteristics.

**Results**. Our results showed that: (1) a total of 60,795 nematodes, representing 80 genera were collected. Total abundance, trophic group abundance, and diversity of nematodes peaked at middle elevations. However, in *M. sieversii*, nematode diversity increased with elevation. (2) The soil food web in the wild fruit forest exhibited a relatively high metabolic rate (nematode channel ratio > 0.5). (3) In *J. cathayensis*, the nematode ecological network was more complex and tightly interconnected at low elevations. whereas in *M. sieversii*, greater network complexity was observed at high elevations. (4) Environmental factors (organic matter, pH, total phosphorus, available potassium, total potassium) significantly affected the composition of nematode trophic groups ($p < 0.05$). Moreover, network complexity was a key factor influencing soil multifunctionality. This study provides a theoretical foundation for long-term monitoring of soil health in Tianshan wild fruit forest.

## INTRODUCTION

The global mountain ecosystems, as biodiversity hotspots, are currently experiencing severe impacts from climate change (*Hagedorn, Gavazov & Alexander, 2019*; *Dainese et al., 2024*). The underground ecological processes and functional maintenance mechanisms of these ecosystems remain to be further explored. As a key component of the mountain ecosystem in Central Asia, the Tianshan wild fruit forest is a natural mixed forest. It is composed of *Juglans cathayensis*, *Malus sieversii*, *Prunus armeniaca*, *Populus tremula*, *Cerasus japonica*, and other wild fruit trees (*Zhang, 2016*). The Tianshan wild fruit forest is not only the origin and genetic pool of cultivated deciduous fruit trees worldwide (*Zhang & Zheng, 2020*), but also one of the key areas of biodiversity in China. Wild fruit forests mainly grow in arid areas, but they also occupy local humid microenvironments, such as riverbanks and sheltered valleys. The unique geographical environment has created the distinctive and important ecological value of this community vegetation (*Nuer, Zhang & Zhang, 2015*). In recent years, large-scale human development, global climate change, and pests have led to widespread degradation of the Tianshan wild fruit forest. This has resulted in severe ecosystem damage and has placed its populations at risk of extinction (*Chen et al., 2018a*).

Studies have shown that there are large spatial differences in the geographical distribution patterns of biodiversity at different scales, such as latitude and elevation (*Li et al., 2021*). The changes of environmental factors such as temperature, humidity and soil on the elevation gradient are significantly more rapid than those on the latitude gradient (*Li & Ma, 2018*). These dramatic shifts in abiotic factors affect the distribution of diversity and the mechanisms of species coexistence among plants, animals, and microorganisms (*Zizk & Antonelli, 2018*). This is of great significance for studying biodiversity and its response to environmental change. Nematodes are the most abundant metazoans in soil (*Kouser, Shah & Rasmann, 2021*). They participate in important ecological processes, such as organic matter decomposition, nutrient mineralization, and regulation of the soil microbial community structure in the underground food web (*Shao & Fu, 2007*). They occupy multiple trophic levels in the food web (*Yeates et al., 1993*) and play a key role in energy flow and ecosystem function assessment of soil food webs (*Chen et al., 2018b*). Therefore, they are considered significant bioindicators for monitoring ecosystem health and environmental change (*Salamun et al., 2017*; *Zhang, Ji & Yang, 2021*). Studies have shown that forest soils at lower elevations tend to have higher nematode diversity and abundance than those at higher elevations, possibly due to more resilient ecosystems at lower elevations (*Yeates, 2007*). Similarly, research on alpine forests in India indicates that the abundance and diversity of nematodes decrease as altitude increases, and high-elevation areas may face greater ecological pressure (*Afzal et al., 2021*). Nematode diversity in the Norikura Mountains of Japan exhibited a unimodal distribution along the elevation gradient (*Dong et al., 2017*). This pattern may be related to the "mid-domain effect" in ecology. The mid-domain effect (MDE) theory posits that if the species in a community are randomly distributed between the top and the base of a mountain, the middle elevations are most likely to have the highest abundance of overlapping species (*Colwell & Lees, 2000*). In summary, the distribution patterns of nematode communities along elevation gradients

are not always the same in different regions. Currently, four typical altitudinal distribution patterns of soil biological communities have been identified: (1) maximum diversity at middle elevations (*Dong et al., 2017*), (2) increasing diversity with elevation (*Kergunteuil et al., 2016*), (3) decreasing diversity with elevation (*Liu et al., 2019*), and (4) no significant relationship with elevation (*Qing et al., 2015*).

Soil multifunctionality (SMF) refers to the ability of soil to simultaneously provide and maintain multiple ecosystem function, such as water regulation, nutrient supply, carbon storage, and maintenance of soil biodiversity (*Creamer et al., 2022*; *Hu et al., 2024*). SMF emphasizes the overall performance of soil in supporting multiple ecosystem functions, rather than focusing only on single or multiple soil functions (*Manning et al., 2018*). This is a complex ecological process, jointly driven by biological factors (such as biodiversity) and abiotic factors (such as global change) (*Zhao & Wang, 2024*). For example, reduced fungal diversity leads to decreased soil multifunctionality in boreal forests, particularly in functions related to nutrient cycling and climate regulation (*Li et al., 2022*). Plant diversity can enhance soil multifunctionality and organic carbon content by increasing the diversity of belowground organisms (*Liu et al., 2022*). In addition, studies have shown that co-occurrence networks between fungal and bacterial communities at low elevations exhibit higher complexity and interconnectedness (*Li et al., 2020*; *Yang et al., 2021*). Moreover, bacterial and fungal communities are closely associated with bacterial-feeding and fungal-feeding nematodes (*Yeates et al., 1993*). In recent years, co-occurrence network analysis has been widely used to explain species interactions and reveal complex connections among organisms. This method is also applied to assess the complexity and stability of ecological networks and to explore ecosystem responses to environmental changes (*Przulj & Malod-Dognin, 2016*; *Yuan et al., 2021*).

Most wild fruit forests exhibit a zonal (or banded) distribution along the vertical axis of mountains (*Zhang & Zheng, 2020*). Short-range changes in elevation can alter climate and soil microenvironments, including primary productivity, soil microbial biomass, and moisture. These factors can affect the distribution of soil nematode communities (*Song et al., 2016*; *Choudhary et al., 2023*). Therefore, it is necessary to conduct an in-depth analysis of the response mechanisms of the underground ecosystem in wild fruit forests to changes in elevation. This study focuses on two typical constructive species in the Tianshan wild fruit forest: *J. cathayensis* and *M. sieversii.* Among these species, *J. cathayensis* is characterized by a deep-root system, with the main root of mature trees reaching depths of up to six m (*Huang et al., 2019*). The litter of *J. cathayensis* decomposes slowly, which facilitates the accumulation of soil organic matter (*She, 1994*). *M. sieversii* has a shallow root system and mainly depends on topsoil resources. Its litter decomposes easily, which promotes the rapid mineralization of soil organic matter (*Rong et al., 2024*). In this study, we examined the abundance, diversity, ecological indices, and co-occurrence networks of nematode communities in the two wild fruit forests along altitude gradients. Then, we analyzed the correlations among environmental factors, soil multifunctionality, and soil nematode communities. The purpose of this study was to elucidate the response and driving mechanism of soil nematode community structure and network complexity to elevation gradient in wild fruit forest. This study is significant for predicting how wild fruit forest

ecosystems respond to global changes. We hypothesized that: (1) Suitable hydrothermal conditions, vegetation coverage, and sufficient nutrient resources at middle elevations jointly drive the highest total abundance and trophic group abundance of soil nematodes. (2) The responses of nematode diversity, ecological indices, and co-occurrence networks to elevation changes differ between the two types of wild fruit forests. (3) Environmental factors at different elevation gradients drive soil multifunctionality by affecting nematode community structure and network complexity.

## MATERIALS & METHODS

### Study area

The study site (82°15′28–82°17′23″E, 43°22′56″–43°25′40″N) was located in the Juglans Cathayensis Nature Reserve, Gongliu County, Xinjiang Uygur Autonomous Region, China (Fig. 1). The reserve mainly consists of four gullies: main, west, south, and east, each of which forms a ''V'' shape. The terrain on both sides of the gullies is steep, with slopes ranging from 30°  to 50° (Dong et al., 2012). The reserve covers approximately 1,180 hm$^2$, and elevation gradually increases from north to south (Zhang, 2016). The reserve experiences a temperate continental climate. The average annual temperature is 7.6 °C. Average annual precipitation is 580 mm, and annual evaporation is 1,200 mm. Rainfall is higher in spring and summer (Zhang, 2016). Abundant local precipitation, a significant winter inversion layer, and topographic features that block cold waves create favorable conditions for the growth and distribution of wild fruit forests (Zhang, 2016). *J. cathayensis* is the dominant species in the study area, with associated trees including *M. sieversii* and *Salix tianschanica*. Understory shrubs include *Spiraea hypericifolia*, *Lonicera altmannii*, and *Berberis heteropoda*. Understory herbs include *Festuca gigantea*, *Urtica fissa*, *Aegopodium alpestre*, and others (Zhang, 2016).

### Sample collection

Because *J. cathayensis* and *M. sieversii* are mainly distributed at elevations between 1,200 and 1,500 m, we established six elevation gradients along the mountain slope: 1,480 m (E1), 1,401 m (E2), 1,351 m (E3), 1,305 m (E4), 1,252 m (E5), and 1,207 m (E6). All plots at the six elevations consisted of *J. cathayensis* and *M. sieversii* forests, and the mixed stands accounted for at least 80% of the total basal area. The basic characteristics of the plots at different elevations are presented in Table 1.

In September 2023, sample plots were established along the elevation gradient on the shady slope of the study area. First, four trees with similar trunk diameter and crown width were randomly selected from each plot, ensuring that each tree was at least 20 m apart. Next, surface litter was removed, and soil from 0 to 20 cm depth was collected from the four cardinal directions (east, south, west, and north) around each tree. Soil samples from the four directions were thoroughly mixed to obtain a composite sample of approximately 200 g. Finally, the collected soil samples were transported on ice and brought back to the laboratory. A total of 96 soil samples were obtained in this study. One portion was used for nematode extraction, while the other was used to determine soil physicochemical properties after drying, grinding, and sieving.

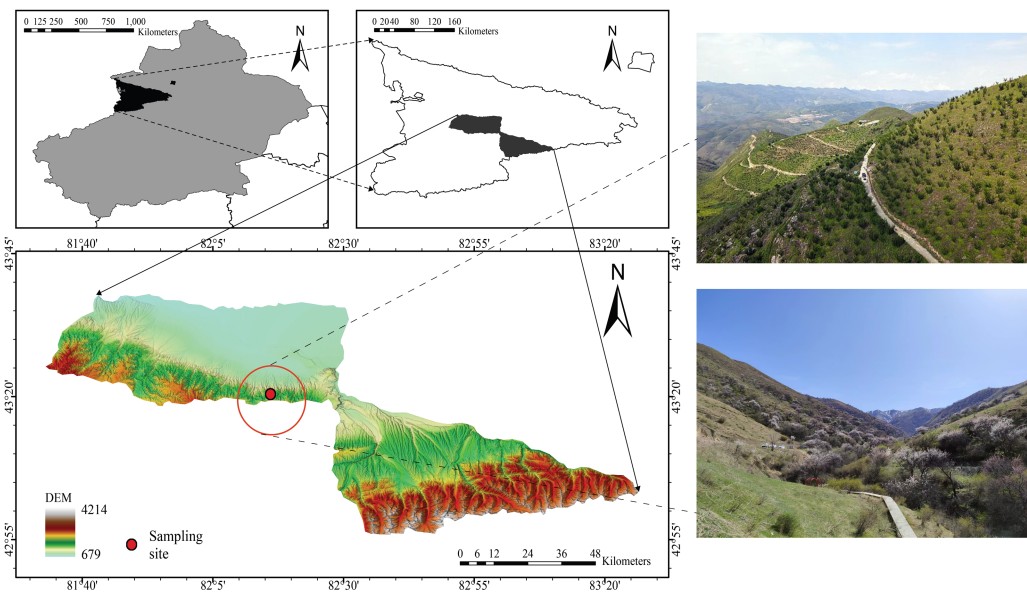

**Figure 1 Location of the study area in the wild fruit forests of Tianshan Mountain.**

## Determination of soil physicochemical characteristics

Soil pH was measured using a FiveEasy Plus pH meter. Electrical conductivity (EC) was determined using a HANNA HI 2315 conductivity meter. Soil total phosphorus (TP) and available phosphorus (AP) were determined by the molybdenum-antimony anti-colorimetric method (*Bao, 2008*). Similarly, total potassium (TK) and available potassium (AK) contents were determined by atomic absorption spectrometry (*Bao, 2008*). Total nitrogen (TN) was measured using the perchloric acid-sulfuric acid digestion method. Soil organic matter (OM) was determined by the potassium dichromate volumetric method (*Bao, 2008*). First, a potassium dichromate-sulfuric acid solution was added to the soil sample, thoroughly mixed, and heated to boiling to ensure complete oxidation of organic matter. Next, the remaining potassium dichromate was titrated with a ferrous sulfate solution. Finally, the organic matter content was calculated based on the amount of potassium dichromate consumed. For nitrate nitrogen (NN) and ammonium nitrogen (AN) determination, soil samples were mixed with 0.01 M calcium chloride solution, shaken vigorously, and left to stand for 5 min before filtration. The resulting clear filtrate was analyzed using a flow analyzer (*Bao, 2008*).

## Nematode isolation and calculation

Soil nematodes were isolated by sucrose centrifugation method (454 g/L sucrose solution, centrifuged at 2,000 r/min for 30 s) (*Jenkins, 1964*). The procedure was as follows: 50 g of fresh soil was added to 2,000 ml of distilled water, stirred thoroughly to form a suspension, and allowed to stand for 1 min. The suspension was filtered through stacked 0.25 mm and 0.038 mm sieves, and the filtrate from the lower sieve was collected. The filtrate was transferred to a centrifuge tube and centrifuged at 2,000 rpm for 2 min. The supernatant was then discarded. Next, 35 ml of 454 g/L sucrose solution was added to the centrifuge

**Table 1  Overview of the sampling plots.**

| Plot code | Elevation (m) | Site | Soil type |
|---|---|---|---|
| E1 | 1,480 | 82.278°E, 43.345°N | Above an elevation of 1,250 m, it is a semi humid grassland with soil composed of black calcareous soil (*Zhang, 2016*). |
| E2 | 1,401 | 82.277°E, 43.346°N | |
| E3 | 1,351 | 82.276°E, 43.346°N | |
| E4 | 1,305 | 82.276°E, 43.346°N | |
| E5 | 1,252 | 82.274°E, 43.349°N | At an elevation of 1,150–1,250 m, it is a semi-arid grassland with chestnut soil (*Zhang, 2016*). |
| E6 | 1,207 | 82.273°E, 43.350°N | |

tube, thoroughly mixed, and centrifuged again at 2,000 rpm for 1 min. The mixed solution of the upper layer of sucrose and nematodes was poured into a 0.038 mm mesh sieve, diluted with water and collected into a 50 ml test tube. The samples were left to stand for 24 h to allow soil nematodes to fully settle. The supernatant was removed with a pipette, and the nematode suspension at the bottom was heated in a 60 °C water bath to inactivate the nematodes. After cooling, TAF fixative was added to preserve the nematodes.

The total number of nematodes in 50 g of fresh soil was converted to the number per 100 g of dry soil based on the measured soil moisture content. Under a light microscope, the first 100 nematodes were identified to the genus level, and their numbers were recorded. If fewer than 100 nematodes were present, all individuals were identified and counted. The identified nematodes were classified into four trophic groups: bacterial-feeding (BF), fungal-feeding (FF), plant-feeding (PF), and omnivorous-predatory (OP), based on their feeding habits and esophageal characteristics (*Yeates et al., 1993*). The isolation and identification of nematodes referred to the book "*Nematodes of the Netherlands*" (*Bongers, 1994*).

Nematode community diversity and functional structure were analyzed by the following indicators:

$$\text{Shannon-Wiener index}: H^{'} = -\sum_{I=1}^{S} n_i/N \times Ln(n_i/N) \tag{1}$$

$$\text{Pielou evenness index}: J' = H^{'}/LnS \tag{2}$$

$$\text{Margalef index}: SR = S - 1/LnN \tag{3}$$

$$\text{Simpson index}: \lambda = \sum (n_i/N)^2. \tag{4}$$

$n_i$ is the number of individuals in the group $i$, N is the total number of individuals, and $S$ is the number of groups in the community.

$$\text{Free-living nematode maturity index/ Plant parasite index}: MI/PPI = \sum_{i=1}^{n} v(i) \times f(i) \tag{5}$$

$v(i)$ is the c-p (colonizer-persister) value of group $i$ of non-plant-feeding (plant-feeding); $n$ is the number of nematode groups. $f(i)$ is the proportion of the number of individuals

in the group $i$ of non-plant-feeding (plant-feeding) to the total number of individuals in all groups of the community.

$$\text{Nematode channel ratio}: NCR = BF/(BF+FF). \tag{6}$$

BF and FF represent the relative abundances of bacterial-feeding and fungal-feeding nematodes, respectively, within the total nematode population. The value of NCR ranges from 0 to 1. A value of 0 indicates complete fungal dominance, while a value of 1 indicates complete bacterial dominance.

$$\text{Enrichment index}: EI = 100 \times (e/(e+b)) \tag{7}$$

$$\text{Structure index}: SI = I00 \times (s/(s+b)). \tag{8}$$

$s$ represents the structural component of the food web. It mainly refers to bacterial- and fungal-feeding groups with cp values of 3 to 5, as well as omnivorous-predatory nematodes with cp values of 2 to 5; $e$ represents the enrichment component of the food web. It mainly refers to bacterial-feeding nematodes with a cp value of 1 and fungal-feeding nematodes with a cp value of 2; $b$ represents the basal component of the food web. It mainly refers to bacterial-feeding and fungal-feeding nematodes with a cp value of 2. Nematode fauna analysis based on the enrichment index (EI) and structure index (SI) can reflect soil nutrient enrichment and changes in the food web (*Ferris, Bongers & De Goede, 2001*; *Berkelmans et al., 2003*).

## Calculation of soil multifunctionality

We selected eight soil indicators (SOC, TN, AN, NN, TP, AP, TK, AK) to characterize soil multifunctionality, and calculated soil multifunctionality using the average method. SOC reflects the carbon cycle in the ecosystem. AN, NN, and TN are indicators of nitrogen cycling. AP and TP together represent phosphorus cycle process. TK and AK together reflect the storage and supply status of potassium in the soil. The average method involves standardizing all indicators by converting them to a 0–1 scale, followed by calculation of $Z$ scores for each standardized indicator. The mean value of all $Z$ scores reflects the average level of soil function, that is, soil multifunctionality (*Maestre et al., 2012*).

## Statistical analysis

Data analysis and plotting were performed using SPSS 26.0 (IBM Corp., Armonk, NY, USA) and R 4.1.3. One-Way Analysis of Variance (ANOVA) was used to test differences in soil nematode abundance, diversity, and ecological indices among different elevation gradients. Duncan method was used for multiple comparisons of the post-test. The Kruskal–Wallis $H$ test was used when data did not meet the assumptions of normality and homogeneity of variance required for ANOVA.

Principal coordinate analysis (PCoA) based on the R package "ape" was used to explore similarities in nematode community structure across different elevation gradients. The significance of group differences was evaluated using the Adonis test. To explore potential interactions among nematode genera, the "psych" package in R was used to calculate species correlations. Nematode co-occurrence networks were constructed separately for
each elevation level, which reduced the influence of elevation on network construction. Network visualization was performed by Gephi software. Topological parameters, including average degree, average clustering coefficient, density, and modularity, were calculated using the R package "igraph" to reflect nematode network complexity. The Mantel test, conducted using the R package "gcor", was used to explore the correlation between environmental factors and the composition of nematode trophic groups. A random forest model was used to assess the importance of multiple environmental factors for nematode diversity, using the "randomForest" and "rfPermute" packages in R. The relationships among elevation, environmental factors, nematode community, network complexity, and soil multifunctionality were analyzed using a partial least squares path model (PLS-PM) constructed with the R package "plspm".

## RESULTS

### Nematode community composition and abundanc

A total of 60,795 nematodes representing 80 genera were collected. The average density was 1,158 nematodes per 100 g of dry soil. In the *J. cathayensis* forest, the dominant nematode genera were *Paratylenchus*, *Acrobeles*, and *Aphelenchus*. *Paratylenchus* showed the highest relative abundance at E5 (23.11%). *Acrobeles* was the dominant genus at E1 (17.52%) and E4 (12.43%). *Aphelenchus* was mainly enriched in low-elevation soils (E6), accounting for 10.85% of the total nematode population at E6 (Fig. 2A, Table S1).

In the *M. sieversii* forest, no single dominant genus was found at E1, and the remaining 25 common genera accounted for 88.99% of the total nematode population at E1. *Paratylenchus* was the common dominant genus at E2, E3, E5, and E6, accounting for 21.64% at low elevation (E5). Additionally, the dominant genera at E2 were *Coslenchus* (10.95%) and *Basiria* (14.11%). *Acrobeloides* was mainly enriched in soils at E4 (10.19%). *Helicotylenchus* was the dominant genus at E4 (13.49%) and E6 (10.42%) (Fig. 2A, Table S1).

At all elevations in both wild fruit forests, the relative abundance of trophic groups followed the order: plant-feeding >bacterial-feeding >fungal-feeding >omnivorous-predatory (Fig. 2B). In *J. cathayensis*, the total abundance of nematodes, as well as the abundance of trophic groups (bacterial-feeding, fungal-feeding, plant-feeding), was the highest at middle elevation (E3, E4) (Fig. 2C). However, in *M. sieversii*, the total abundance of nematodes peaked at E2. The abundance of bacterial-feeding and fungal-feeding nematodes was the highest at middle elevation (E3, E4), whereas the abundance of plant-feeding nematodes was the highest at E5 (Fig. 2C). Based on the principal coordinate analysis (PCoA) analysis, the nematode community structure of E2, E3, and E4 were similar in the *J. cathayensis* (Fig. 3A). In contrast, in *M. sieversii*, there were significant differences in nematode community composition between E4 and E5 (Fig. 3B).

### Nematode diversity and ecological index

In *J. cathayensis*, the Shannon–Wiener index ($H^{'}$) and Margalef index (*SR*) at the middle elevation (E3) were significantly higher than those at high elevation (E1) ($p < 0.05$). The maturity index (MI) of nematodes at E6 was significantly higher than at E1 and E3. The enrichment index (EI) was significantly higher at E3 than at E4. The structure index (SI)

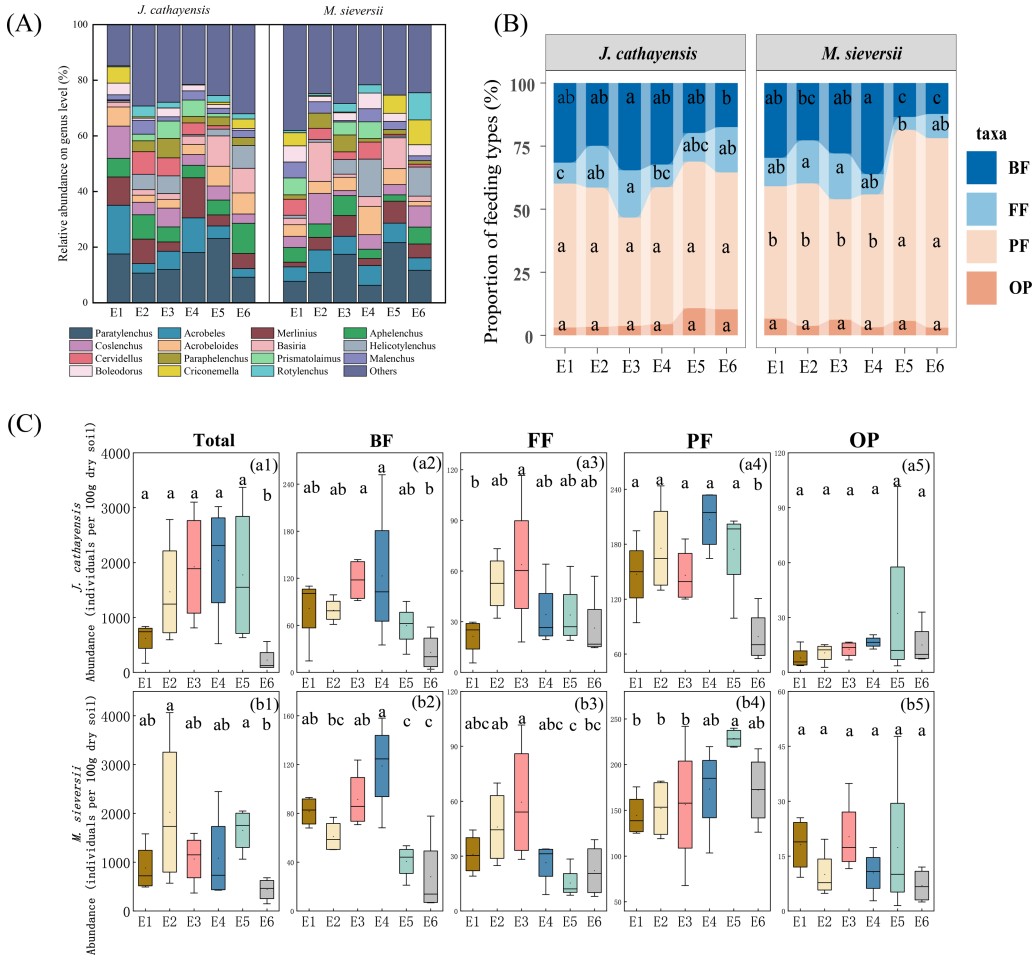

**Figure 2** **The composition of nematode communities at different elevation gradients in two Tianshan wild fruit forests.** (A) the relative abundance of the top 15 most abundant nematode genera, (B) the proportion of relative abundance of nematodes trophic group, and (C) changes in the abundance of trophic group of nematodes along the elevation gradient. Different lowercase letters indicated significant differences in the elevation gradient ($p < 0.05$).

generally increased with the decrease of elevation, and SI was significantly higher at E6 than at E1 ($p < 0.05$). The nematode channel ratio (NCR) was significantly lower at E6 than at the other five elevations ($p < 0.05$) (Table 2).

In *M. sieversii*, the $H'$ and *SR* indices at E1 were significantly higher than those at E5 ($p < 0.05$) (Table 2). The plant parasite index (PPI) was the highest at E6 and the lowest at E2. Contrary, NCR had its lowest value at E6 (Table 2). The nematode food web diagnostic based on EI-SI showed that all elevations in the two wild fruit forests generally fell into quadrant C (poor soil nutrients, low disturbance) and quadrant D (poor soil nutrients, high disturbance). Only E3 and E6 in *J. cathayensis* were located in quadrant B, indicating that the food web at these elevations had good soil nutrient conditions and was less disturbed (Fig. 4).

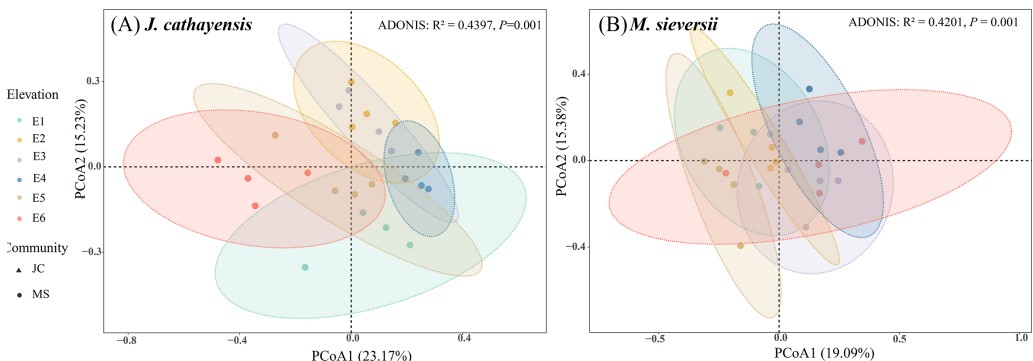

**Figure 3** Principal coordinate analysis (PCoA) of nematode communities in *J. cathayensis* forest (A) and *M. sieversii* forest (B).

## The complexity of nematode ecological network

We constructed nematode co-occurrence networks at different elevation gradients in two wild fruit forests under the same threshold ($r > 0.8$, $p < 0.01$) (Fig. 5). In the nematode network of *J. cathayensis*, analysis along the elevation gradient showed that the topological parameters indicating network complexity, namely, average degree, edge, density, and average clustering coefficient, were higher at low elevation (E6) than at other elevations (Table 3). However, in the ecological network of *M. sieversii*, the average degree first decreased and then increased with increasing elevation (Table 3). These results indicated that the nematode network in the *J. cathayensis* forest had more complex and close relationships at low elevations, while the nematode co-occurrence network in the *M. sieversii forest* was more complicated at high elevations.

## Relationship between soil nematode community and environmental factors

### Relationship between environmental factors with nematode community composition and diversity

The Mantel test showed that soil organic matter (OM), total potassium (TK), and available nitrogen (AN) were significantly correlated with the composition of bacterial-feeding nematodes (BF) in *J. cathayensis*. The composition of fungal-feeding nematodes (FF) was influenced by OM and AN. The composition of omnivorous-predatory nematodes (OP) was significantly correlated with total phosphorus (TP) ($p < 0.05$) (Fig. 6A). In *M. sieversii*, BF was closely associated with soil physicochemical properties, particularly soil pH ($p < 0.01$). FF was significantly regulated by available potassium (AK), and OP was significantly affected by total potassium (TK) ($p < 0.05$, Fig. 6B). Random forest analysis indicated that total phosphorus (TP) was a significant factor influencing nematode diversity in *M. sieversii* (Fig. 6B).

Zhang et al. (2025), *PeerJ*, DOI 10.7717/peerj.20090

**Table 2  Ecological index of soil nematodes at different elevation gradients (mean ± standard error).**

| Habitat | Ecological index | E1 | E2 | E3 | E4 | E5 | E6 |
|---|---|---|---|---|---|---|---|
| *J. cathayensis* | $H'$ | 2.34 ± 0.19c | 2.85 ± 0.03ab | 2.91 ± 0.18a | 2.60 ± 0.17abc | 2.52 ± 0.27bc | 2.64 ± 0.25abc |
| | $J'$ | 0.55 ± 0.12a | 0.58 ± 0.03a | 0.59 ± 0.04a | 0.51 ± 0.06a | 0.52 ± 0.10a | 0.65 ± 0.08a |
| | $\lambda$ | 0.13 ± 0.035a | 0.09 ± 0.01a | 0.08 ± 0.02a | 0.11 ± 0.03a | 0.13 ± 0.05a | 0.10 ± 0.02a |
| | SR | 3.43 ± 0.47b | 5.04 ± 0.21a | 5.29 ± 0.76a | 4.38 ± 0.25ab | 4.16 ± 0.56ab | 4.44 ± 1.17ab |
| | EI | 31.52 ± 6.08bc | 40.09 ± 13.94abc | 56.27 ± 13.38a | 22.92 ± 10.09c | 46.75 ± 16.56abc | 52.83 ± 14.58ab |
| | SI | 28.67 ± 11.66b | 46.09 ± 19.42ab | 53.03 ± 6.51ab | 57.60 ± 7.80ab | 53.67 ± 25.51ab | 65.08 ± 18.58a |
| | MMI | 2.27 ± 0.11b | 2.43 ± 0.17ab | 2.27 ± 0.11b | 2.40 ± 0.04ab | 2.35 ± 0.16ab | 2.55 ± 0.12a |
| | MI | 2.13 ± 0.10b | 2.30 ± 0.18ab | 2.18 ± 0.08b | 2.48 ± 0.13ab | 2.43 ± 0.33ab | 2.62 ± 0.27a |
| | PPI | 2.35 ± 0.12a | 2.51 ± 0.20a | 2.40 ± 0.20a | 2.35 ± 0.07a | 2.31 ± 0.19a | 2.51 ± 0.15a |
| | NCR | 0.78 ± 0.04a | 0.60 ± 0.09ab | 0.66 ± 0.16a | 0.75 ± 0.120a | 0.63 ± 0.07ab | 0.44 ± 0.16b |
| *M. sieversii* | $H'$ | 3.07 ± 0.07a | 2.78 ± 0.23ab | 2.71 ± 0.35ab | 2.78 ± 0.14ab | 2.41 ± 0.46b | 2.55 ± 0.35ab |
| | $J'$ | 0.61 ± 0.02a | 0.55 ± 0.08a | 0.66 ± 0.11a | 0.59 ± 0.02a | 0.59 ± 0.11a | 0.51 ± 0.10a |
| | $\lambda$ | 0.06 ± 0.01a | 0.09 ± 0.02a | 0.10 ± 0.02a | 0.09 ± 0.01a | 0.15 ± 0.10a | 0.14 ± 0.08a |
| | SR | 6.12 ± 0.36a | 5.16 ± 0.65ab | 4.26 ± 1.58bc | 4.61 ± 0.55abc | 3.30 ± 0.78c | 4.60 ± 0.55abc |
| | EI | 41.20 ± 15.84a | 39.91 ± 13.18a | 45.53 ± 16.31a | 31.26 ± 13.41a | 21.09 ± 5.30a | 44.43 ± 13.38a |
| | SI | 64.83 ± 9.09a | 47.55 ± 13.99a | 59.73 ± 13.39a | 53.69 ± 10.08a | 43.94 ± 19.66a | 47.99 ± 21.06a |
| | MMI | 2.46 ± 0.13ab | 2.25 ± 0.15b | 2.42 ± 0.19ab | 2.40 ± 0.08ab | 2.41 ± 0.08ab | 2.55 ± 0.22a |
| | MI | 2.49 ± 0.24a | 2.32 ± 0.17a | 2.42 ± 0.23a | 2.33 ± 0.09a | 2.38 ± 0.24a | 2.41 ± 0.46a |
| | PPI | 2.42 ± 0.14ab | 2.19 ± 0.13b | 2.44 ± 0.16ab | 2.46 ± 0.14a | 2.40 ± 0.08ab | 2.60 ± 0.19a |
| | NCR | 0.73 ± 0.08ab | 0.58 ± 0.10bc | 0.63 ± 0.07bc | 0.83 ± 0.04a | 0.73 ± 0.08ab | 0.48 ± 0.11c |

**Notes.**

Significant *p*-values are shown in bold ($p < 0.05$).

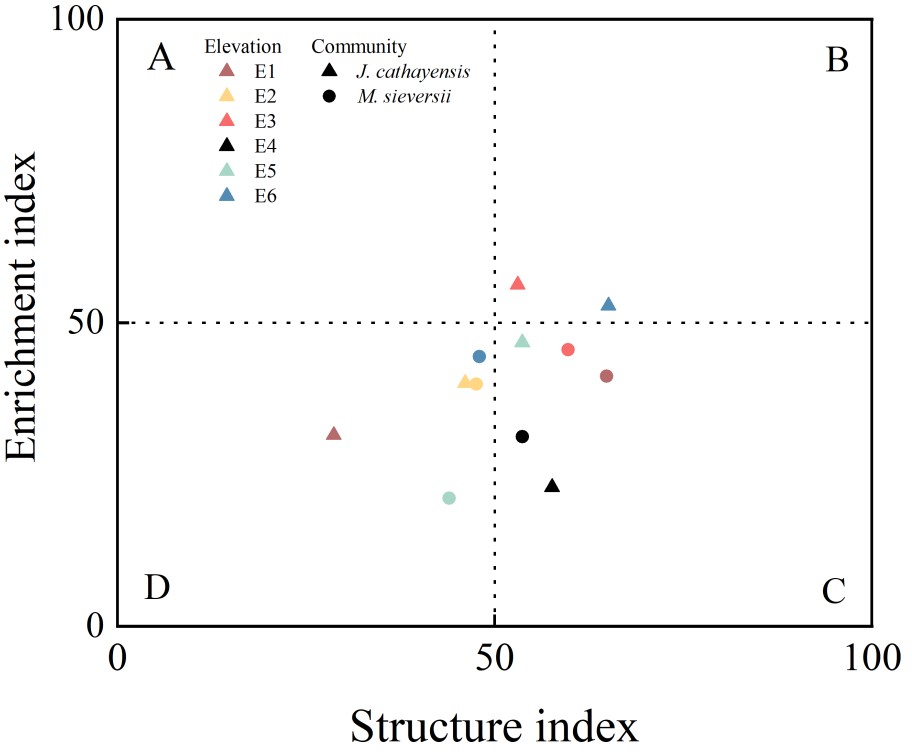

**Figure 4  Food web diagnostic of wild fruit forest at different elevations.**

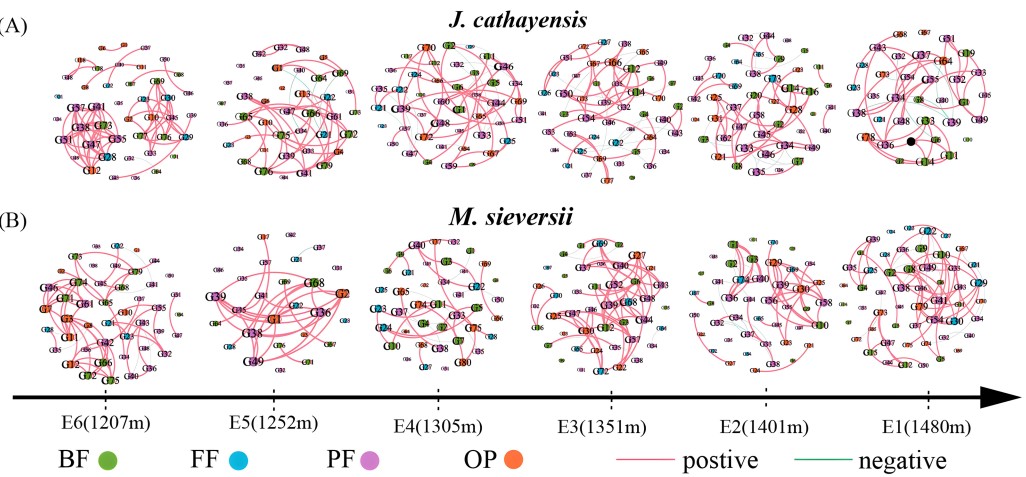

**Figure 5  Nematode co-occurrence networks at different elevation gradients in *J. cathayensis* forest (A) and *M. sieversii* forest (B) in Tianshan Mountains.** Nodes represent nematode genera, node size is proportional to the relative abundance of nematode genera, and edges represent significant correlations between nematode genera.

**Table 3  Topological parameters of nematode co-occurrence networks in wild fruit forests along an altitudinal gradient.**

| Habitat | Stochastic network | E1 | E2 | E3 | E4 | E5 | E6 |
|---|---|---|---|---|---|---|---|
| *J. cathayensis* | Number of node | 35 | 46 | 47 | 44 | 40 | 43 |
| | Number of edge | 72 | 99 | 90 | 92 | 78 | 110 |
| | Average degree | 4.114 | 4.304 | 3.830 | 4.182 | 3.900 | 5.116 |
| | Density | 0.121 | 0.096 | 0.083 | 0.097 | 0.100 | 0.122 |
| | Average clustering coefficient | 0.796 | 0.832 | 0.829 | 0.822 | 0.803 | 0.849 |
| | Modularity | 0.93 | 1.208 | 1.14 | 1.163 | 0.913 | 0.643 |
| *M. sieversii* | Number of node | 52 | 44 | 44 | 41 | 28 | 45 |
| | Number of edge | 112 | 92 | 9 | 67 | 57 | 104 |
| | Average degree | 4.622 | 4.182 | 4.500 | 3.268 | 4.071 | 4.226 |
| | Density | 0.105 | 0.097 | 0.105 | 0.082 | 0.151 | 0.081 |
| | Average clustering coefficient | 0.841 | 0.841 | 0.844 | 0.782 | 0.805 | 0.85 |
| | Modularity | 1.086 | 0.971 | 0.873 | 1.274 | 0.414 | 0.898 |

### Relationship between nematode community, network complexity and soil multifunctionality

The soil multifunctionality of *J. cathayensis* peaked at E3, while that of *M. sieversii* was the highest at E6 (Fig. 7A). To improve the accuracy of the model, only environmental variables significantly related to nematode community structure were included in the PLS-PM model. The goodness of fit was 0.50 for the *J. cathayensis* model and 0.59 for the *M. sieversii* model. The PLS-PM model showed that the elevation gradient could directly affect soil multifunctionality. It could also indirectly influence soil multifunctionality by altering soil properties, nematode abundance, and network complexity (Fig. 7B). In *J. cathayensis*, soil properties (OM, TK, TP, AN) had the greatest influence on soil multifunctionality. In *M. sieversii*, nematode abundance and network complexity were the key factors affecting soil multifunctionality (Fig. 7C).

## DISCUSSION

### Effects of elevation change on nematode community composition in Tianshan wild fruit forest

Soil nematodes serve as important biota indicating environmental conditions, and the geographic distribution of nematodes reflects changing patterns in the geographic environment to a certain extent (*Zhong, Zhang & Hou, 2022*). Elevation change represents a comprehensive change in physical factors such as moisture and temperature (*Zhong, Zhang & Hou, 2022*). Vegetation composition, soil nutrients, and climate variables change greatly with elevation (*Yu et al., 2021*; *Kashyap et al., 2022*). Therefore, elevation changes inevitably lead to shifts in the distribution of nematode communities. However, it is still unclear how much altitude difference is needed to cause significant changes in the nematode community, which is a question worthy of further investigation. The PCoA results of this study show that the composition of nematode communities is more similar

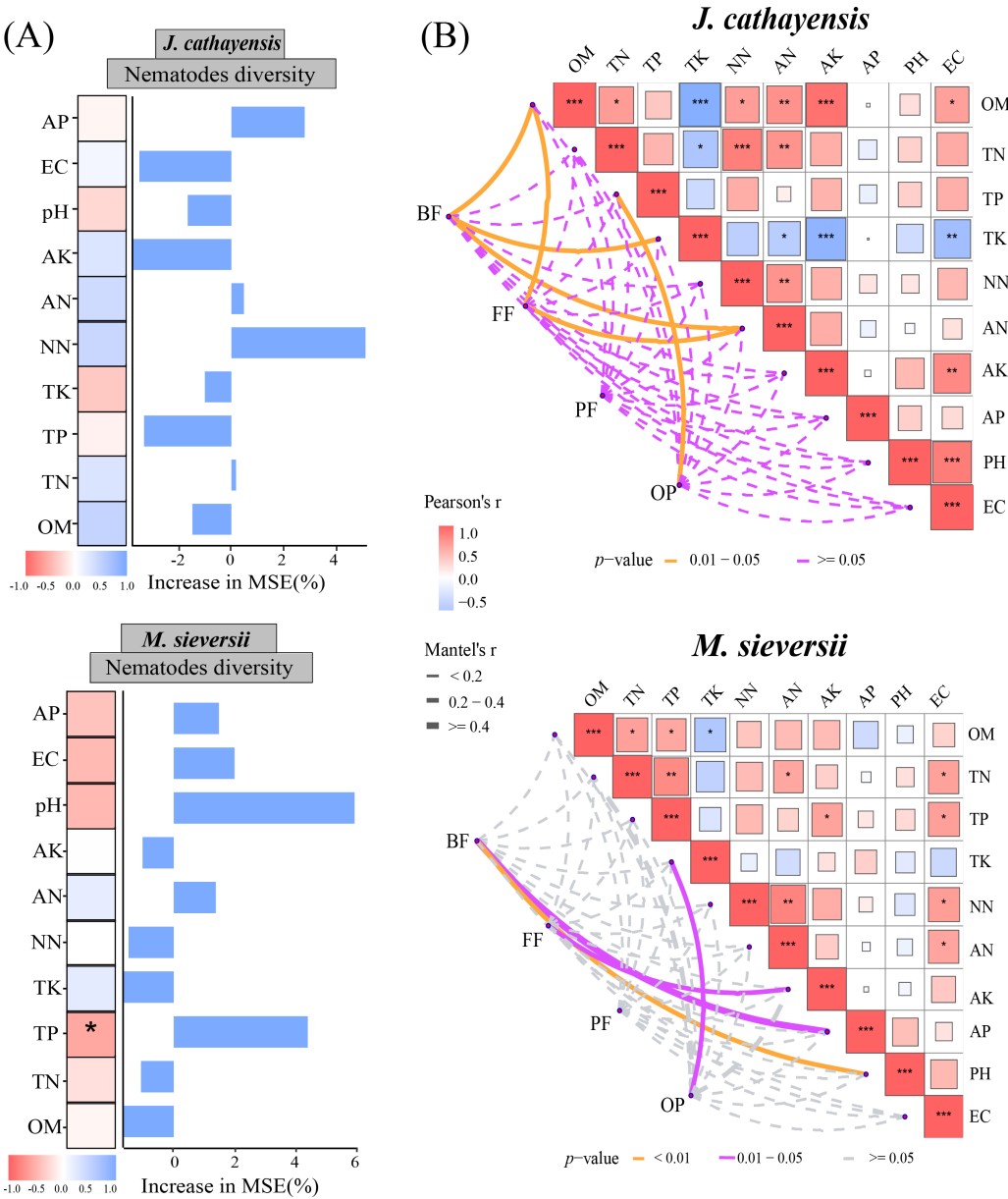

**Figure 6** **Effects of soil physicochemical properties on nematode communities.** (A) Correlation analysis between the composition of nematode trophic groups and environmental factors based on the Mantel test. (B) Random forest analysis to predict the importance of environmental factors on nematode diversity. OM, Soil organic matter; TN, Soil total nitrogen; TP, Soil total phosphorus; TK, Total potassium; NN, Nitrate nitrogen; AN, Ammonium nitrogen; AK, Available potassium; AP, Available phosphorus; pH, Soil pH; EC, Electrical conductivity; BF, composition of bacterial-feeding nematodes; FF, composition of fungal-feeding nematodes; PF, composition of plant-feeding nematodes; OP, composition of omnivorous-predatory nematodes.

among elevations of 1,480 m and below 1,250 m, while nematode communities at middle elevations (1,305–1,351 m) display stronger specificity compared to other elevations. These results indicate that, even though the Tianshan wild fruit forest spans an elevation range

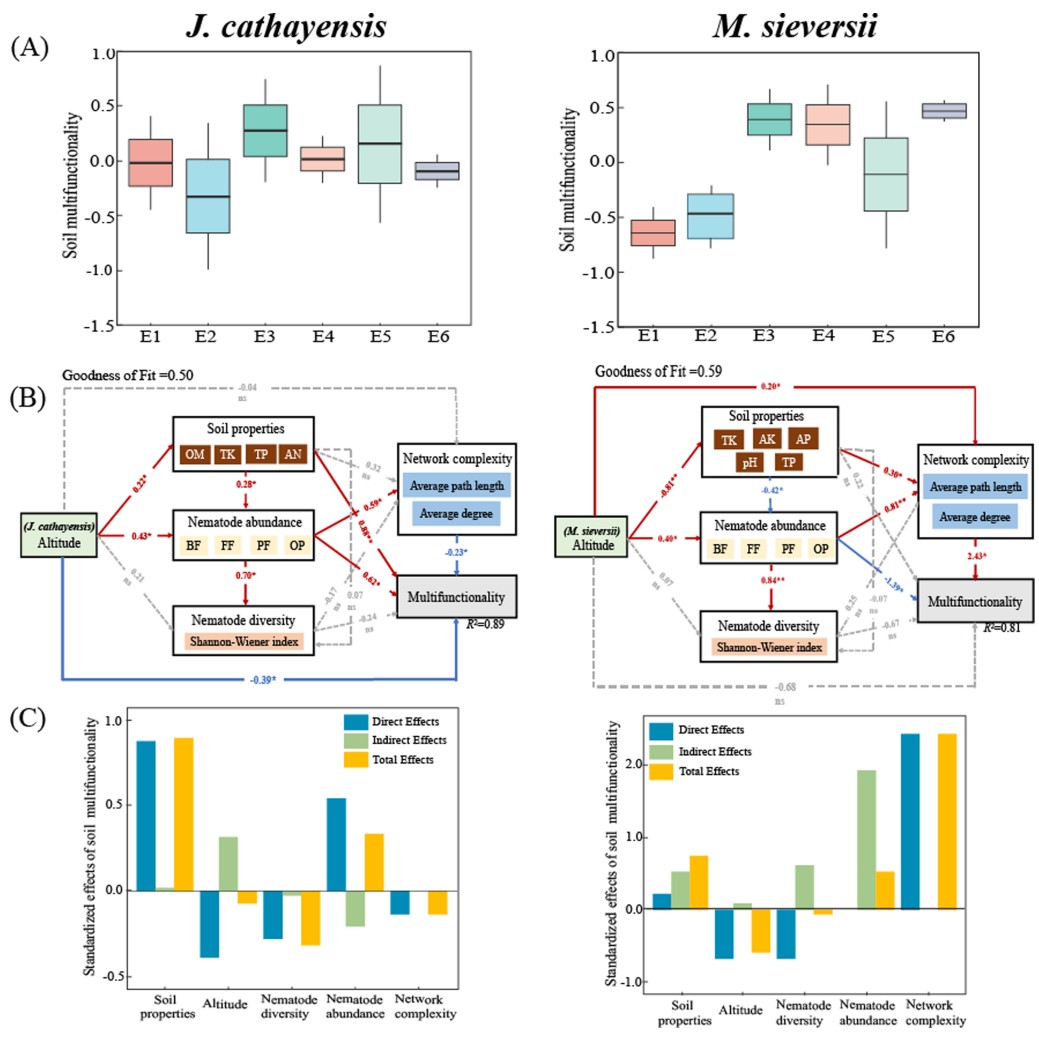

**Figure 7** **Driving factors of soil multifunctionality.** (A) Soil multifunctionality under different elevation gradients, (B) partial least squares path model (PLS-PM), and (C) standardization effect revealed the pathways of elevation gradient on soil multifunctionality. Standardized path coefficients were given near the arrow. *, $p < 0.05$; **, $p < 0.01$.

of only about 300 m, there are still obvious differences in the distribution pattern of nematode communities within this narrow gradient. We observe that the total abundance of nematodes and the abundance of trophic groups (bacterial-feeding, fungal-feeding, plant-feeding) peak at the middle elevations in both wild fruit forests. This phenomenon is consistent with our first hypothesis. It also aligns with the classic "mid-domain effect" and the "optimal environment hypothesis" of mountain ecosystems. First, the soil moisture content, SOC, TN, NN, and AN in the Tianshan wild fruit forest peak at middle elevations (Table S2). These conditions provide an optimal ecological environment for the survival and reproduction of nematode communities. Furthermore, the survey of wild fruit forests in the Tacheng area of China found that both tree species frequency and niche breadth are the highest at middle elevations (*Lu et al., 2024*). In each mountain steppe, there is

a positive correlation between precipitation and species richness (*McCain, 2007*). More precipitation and lower temperatures at middle elevations increase soil moisture and porosity. These conditions are more conducive to nematode feeding activity and promote higher nematode abundance and diversity (*Dong et al., 2017*). Studies in the Andes and the European Alps also show a similar mid-elevation peak in the abundance of soil fauna and microorganisms along the elevation gradient (*Nottingham et al., 2018*). Uniform resource distribution, moderate environmental stress, and habitat heterogeneity are considered core factors driving the mid-elevation peak distribution (*Sundqvist, Sanders & Wardle, 2013*). In summary, the mid-elevation peak in nematode abundance in the Tianshan wild fruit forest is jointly determined by favorable environmental gradients, vegetation coverage, and geographical constraints at both the summit and the base of the mountain. This distribution pattern also indicates that, under moderate stress, an adequate supply of resources can effectively promote biomass accumulation.

Among nematode trophic groups, plant-feeding nematodes mainly feed on plant leaf litter and roots. They respond most directly and rapidly to changes in plant communities (*Mcsorley, 1997*). Therefore, plant-feeding nematodes are more likely to inhabit mid-elevation soils where wild fruit forests are more distributed. The abundance of omnivorous-predatory nematodes does not significantly relate to elevation change, which is consistent with the findings of *Kergunteuil et al. (2016)*. This pattern may be related to the higher trophic position of these nematodes in the soil food web. Omnivorous-predatory nematodes can flexibly select different prey in response to varying environmental conditions (*Afzal et al., 2021*). In addition, our study finds that nematode richness in the Tianshan wild fruit forest (80 genera) is significantly higher than in the Changbai Mountain North Slope forest (60 genera) (*Tong, Xiao & Wang, 2009*) and Yunnan Ailao Mountain (51 genera) (*Li, 2008*). The high nematode richness in wild fruit forests may be attributed to their evolutionary history and unique geographical environment. The climate of the Tianshan wild fruit forest is complex, and the vegetation and soil types are abundant (*Zhang, 2016*). This favorable natural environment provides abundant food resources and ample living space for diverse nematode taxa.

## Effect of elevation change on ecological function of nematode community

To understand important information of soil food web stability and decomposition pathways (*Ferris, Bongers & De Goede, 2001*), several critical functional and community indices were selected in this study. The nematode channel ratio (NCR) reflects the main channels of energy flow in the soil food web and assesses the relative importance of bacterial and fungal channels in organic matter decomposition (*Yeates, 2003*; *Liu et al., 2016*). Our results show that, except at 1,207 m, the overall catabolic process of the soil food web was faster at other elevations in the wild fruit forest (NCR > 0.5). The soil food web was more inclined toward bacterial channels with higher organic matter content. This finding differs from the North Carolina forest, where decomposition is dominated by fungal pathways (*Neher et al., 2005*). This difference may be attributed to regional soil moisture conditions. Fungi are aerobic microorganisms, and the high soil moisture in the Tianshan wild fruit

forest often leads to hypoxic conditions, which are unfavorable for the proliferation of fungal communities. As a result, fungal-feeding nematodes are inhibited, and bacterial decomposition becomes dominant. The free-living nematode maturity index (MI) and plant parasite index (PPI) can reflect the succession status of nematode communities and the degree of disturbance in soil ecosystems (*Bongers & Ferris, 1999*; *Yeates, 2003*). The lower MI indicates the earlier successional stage, higher frequency of environmental disturbances, and lower ecosystem stability (*Bongers & Ferris, 1999*; *Yeates, 2003*). In our study, nematode food web diagnostic shows that the low and middle elevations of *J. cathayensis* are less disturbed. The lowest MI at E6 and the more complex network at low elevations support this conclusion. In general, greater ecological network complexity is associated with richer biodiversity, closer species interactions, stronger community resistance to external disturbance, and higher ecosystem stability (*Landi et al., 2018*). The structure index (SI) decreases with increasing elevation in *J. cathayensis*. This trend is consistent with results from the Pir-Panjal Mountains (*Afzal et al., 2021*), and reflects the low complexity, loose structure, and reduced connectivity of the soil food web at high elevations in *J. cathayensis*. In *M. sieversii*, the MI, SI, and EI index are not affected by elevation, indicating that the soil nematode community structure of *M.sieversii* had high stability. *Kergunteuil et al. (2016)* also observes similar results. The PPI index is positively correlated with the frequency of disturbance (*Ruess, 2003*). The PPI index of *M. sieversii* is lowest at high elevation (1,401 m), indicating that the nematode community at high elevations of *M. sieversii* is less subject to environmental stress and more stable. Studies have also shown that MI is significantly positively correlated with AN, and an appropriate increase in nitrogen content is conducive to improving the anti-interference ability of the ecosystem (*Zhang et al., 2016*).

In addition, we found that although the patterns of nematode abundance in the two wild fruit forests were similar along the elevation gradient, the responses of nematode diversity differed significantly. Tree species have a strong influence on the ecological distribution and composition of nematode communities (*Cesarz et al., 2013*), and plant communities can regulate soil nematode community structure through bottom-up effects (*Eisenhauer et al., 2013*). Specifically, the Shannon–Wiener index (H') and Margalef index (SR) of the nematode community reached their highest values at middle elevations in the *J. cathayensis* forest (Table 2). This may be because *J. cathayensis*, as a deep-rooted species, can efficiently utilize soil moisture and nutrients at resource-rich intermediate elevations (*Huang et al., 2019*). Its well-developed roots not only enhance resource acquisition but also promote the formation of heterogeneous soil microenvironments. This trait helps balance the dual constraints of grazing disturbance at low elevations and low-temperature stress at high elevations, thereby promoting nematode community diversity. In *M. sieversii*, nematode diversity increases with elevation. This may be because high-elevation areas experience less disturbance and have a more stable ecological environment, which is conducive to the survival of rare and endemic nematode species. Furthermore, the low temperatures at high elevation provide a living space for cold-tolerant and adaptable nematodes. In addition, the shallow roots of *M. sieversii* and easily decomposable litter facilitate rapid mineralization of organic matter and nutrient supply at high elevations (*Rong et al., 2024*).

This differentiated pattern of nematode diversity suggests significant differences in the ecological processes driving nematode community composition in the Tianshan wild fruit forest. The distribution pattern of nematode diversity in the wild fruit forest also reflects the particularity of $\alpha$ diversity at middle elevations. In other words, the nematode community at middle elevations has higher richness and diversity. However, this particularity may not be strongly representative due to its narrow elevation gap (*Zhong, Zhang & Hou, 2022*). All of the above results regarding nematode ecological indices support our second hypothesis. Together, these findings indicate that significant differences exist in the composition, trophic structure, and energy flow channels of soil nematodes at different elevations in the *J. cathayensis* and *M. sieversii* forests in Xinjiang.

## Driving factors of nematode community

The composition of soil nematode communities varies along the elevation gradient. This variation is likely related to changes in ecological factors such as local environmental moisture, temperature, vegetation communities, and food resources with increasing elevation (*Zhang et al., 2016*). We find that the composition of nematode trophic groups is associated with SOM, TK, AN, TP, pH, AK, and TK contents (Fig. 6). This finding is consistent with previous studies (*Wall, Bardgett & Kelly, 2010*; *Song et al., 2016*; *Liu et al., 2019*). In the *M. sieversii* forest, pH significantly affects the composition of bacterial-feeding nematode communities. This may be related to the effect of pH on bacterial abundance (*Nottingham et al., 2018*). The abundance of fungal-feeding nematodes is positively correlated with AK content. This may be because AK promotes plant growth, increases root secretions, and enhances the activity of fungal communities (*Zhang et al., 2022*). The density of bacterial and fungal communities is also closely related to the abundance of bacterivores and fungivores (*Yeates et al., 1993*). In addition, high potassium content promotes nematode development, while low potassium levels limit the growth and development of certain nematode species (*Guo et al., 2004*).

Our study also shows that nematode community composition is significantly positively correlated with the complexity of the ecological network (Fig. 7B). Our PLS-PM model shows that soil multifunctionality is not only directly affected by the elevation gradient, but also indirectly regulated by soil properties and nematode communities (Fig. 7B). Other studies have also reported that complex interspecific relationships require greater opportunities for species overlap, trait matching, and species coexistence, which leads to the emergence of more species in natural environments. Therefore, these organisms participate in potential interactions (mutualism, competition, antagonism, predation, and parasitism) that form more complex ecological networks (*Chen et al., 2022*). These interactions directly or indirectly influence soil biological communities and jointly regulate soil multifunctionality (*Montoya, Pimm & Sole, 2006*; *Hu et al., 2024*). In conclusion, the community structure of soil nematodes is complex and influenced by many factors. Therefore, further investigation is needed. In future studies of nematode communities, nematode co-occurrence networks and soil multifunctionality indicators should be flexibly applied to comprehensively investigate the single and combined effects of multiple environmental factors on nematode communities. This approach will help clarify the

general distribution patterns and response mechanisms of soil nematode communities along elevation gradients in different regions.

## CONCLUSIONS

Although the Tianshan wild fruit forest spans only a narrow elevation range of about 300 m, elevation changes still had a significant impact on soil nematode community structure. The nematode communities of the two types of wild fruit forests responded to the elevation change by different mechanisms. In the *J. cathayensis* forest, total abundance, trophic group abundance, and diversity of nematode were the highest at middle elevation. The nematode ecological network was more complex and tightly interconnected at low elevations. In the *M. sieversii* forest, nematode diversity increased with elevation, but the pattern was not strictly monotonic. The nematode co-occurrence network was more complex at high elevations. Soil organic matter, pH, total phosphorus, available potassium, and total potassium content significantly affected the composition of nematode trophic groups. The elevation gradient enhances species interactions by affecting environmental factors and nematode community structure, thus indirectly promoting soil multifunctionality. This study deepens our understanding of how soil biological communities respond to global change and provides a basis for the biodiversity conservation of wild fruit forest ecosystems.

### Funding

This work was supported by the Third Xinjiang Scientific Expedition Program (2022xjkk0405) and the Yili Normal University to enhance the comprehensive strength of the discipline special self-discipline key projects (22XKZZ01). The funders had no role in study design, data collection and analysis, decision to publish, or preparation of the manuscript.

### Grant Disclosures

The following grant information was disclosed by the authors:
Third Xinjiang Scientific Expedition Program: 2022xjkk0405.
Yili Normal University: 22XKZZ01.

### Competing Interests

The authors declare there are no competing interests.

### Author Contributions

- Yulu Zhang conceived and designed the experiments, performed the experiments, analyzed the data, prepared figures and/or tables, authored or reviewed drafts of the article, and approved the final draft.
- Mengyu Yang conceived and designed the experiments, prepared figures and/or tables, and approved the final draft.

- Wenxin Liu conceived and designed the experiments, prepared figures and/or tables, and approved the final draft.
- Zhicheng Jiang conceived and designed the experiments, prepared figures and/or tables, and approved the final draft.
- Yang Zhao conceived and designed the experiments, prepared figures and/or tables, and approved the final draft.
- Gaofeng Li conceived and designed the experiments, prepared figures and/or tables, and approved the final draft.
- Jing Cao performed the experiments, prepared figures and/or tables, and approved the final draft.
- Minru Zhang performed the experiments, prepared figures and/or tables, and approved the final draft.
- Haijun Yang conceived and designed the experiments, authored or reviewed drafts of the article, and approved the final draft.
- Dong Cui conceived and designed the experiments, authored or reviewed drafts of the article, and approved the final draft.

### Data Availability

Raw data is available in the Supplementary File.

### Supplemental Information

Supplemental information for this article can be found online at http://dx.doi.org/10.7717/peerj.20090#supplemental-information.

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
