# Peer review of "Response of soil nematode community structure, diversity, and ecological network to elevation gradients in wild fruit forest of Tianshan Mountain, China"

_PeerJ, doi:10.7717/peerj.20090_

## Round 0.1 · original submission · Major Revisions

Dear Dr. Zhang, I want to give you the opportunity to deeply revise this manuscript and clearly support each of the conclusions statistically. One of the reviewers summarized the shortcomings of this manuscript:

1. The experimental design is not clear, particularly with respect to the term “wild fruit forest”.
2. The authors’ choice of elevation gradient (first hypothesis) is not supported by the literature review in the introduction.
3. Supporting soil chemistry data is briefly mentioned in the methods, however, there is no information regarding how these parameters were measured, nor is there a summary data table included.
4. Data interpretation by the authors is insufficient within the context of the overall literature describing changes in nematode ecology, and the authors have elected to draw conclusions that their data cannot support as it is presented.
5. The manuscript would greatly benefit from English language editing services to improve clarity.

I hope that the new version of the article will be approved by the reviewers and can be published.

**Language Note:** The review process has identified that the English language must be improved. PeerJ can provide language editing services - please contact us at [email protected] for pricing (be sure to provide your manuscript number and title). Alternatively, you should make your own arrangements to improve the language quality and provide details in your response letter. – PeerJ Staff

·

Basic reporting

The manuscript is well-structured and follows a logical flow from introduction to conclusion. The abstract provides a concise summary of the study, including objectives, methods, key results, and implications. The introduction effectively sets the context by highlighting the ecological importance of nematodes and the unique characteristics of the Tianshan wild fruit forest. The methods section is detailed, allowing for reproducibility, and the results are presented clearly with appropriate figures and tables. However, some minor improvements could enhance clarity:
Clarify terminology: Some terms (e.g., "nematode channel ratio," "soil multifunctionality") are not explicitly defined in the introduction or methods, which may confuse readers unfamiliar with these metrics.

Experimental design

The experimental design is robust, with six elevation gradients and two forest types (JC and MS) sampled systematically. Key strengths include:
Sampling: Replication (four trees per elevation) and soil sampling depth (0–20 cm) are appropriate for nematode studies.
Nematode analysis: The use of sucrose centrifugation and taxonomic identification aligns with standard practices.

Statistical methods: Advanced techniques (PCoA, Mantel tests, PLS-PM, co-occurrence networks) are well-suited to address the hypotheses.

Validity of the findings

The findings are supported by the data and analyses, but some aspects warrant further scrutiny:
Mid-domain effect - The explanation for mid-elevation peaks in abundance (lines 342–344) is plausible but speculative. Correlating nematode data with plant community metrics (e.g., biomass, diversity) would strengthen this argument.
Network complexity- The interpretation of network metrics (e.g., higher complexity = more stable?) could be expanded. Are there established thresholds for "high" vs "low" complexity in nematode networks?
Soil multifunctionality- The selection of eight soil indicators is justified, but their equal weighting in the index may oversimplify interactions. Discuss potential biases (e.g., if one nutrient dominates multifunctionality).

Additional comments

The novel focus on nematode networks in understudied Tianshan wild fruit forests.
Comprehensive integration of community ecology, network theory, and soil science.
Clear implications for conservation and response to global change.
Areas for improvement:
Hypothesis 1: The results partially support this (mid-elevation peak in JC but not MS). Discuss why MS deviates (e.g., tree species traits, microhabitat differences).
Tree species effects: The contrast between JC and MS is intriguing but underexplored. Include a brief comparison of their litter quality/root exudates, which may drive nematode differences.
Broader context: Link findings to similar studies (e.g., alpine or tropical forests) to highlight universality/unique aspects of this system.

Reviewer 2 ·

Basic reporting

-

Experimental design

More details are needed on the nematode extraction method, especially explaining the procedure to ensure that the 100 identified nematodes were randomly selected from the sample.
Explain whether your data meet the requirements for ANOVA analysis.

Validity of the findings

-

Additional comments

Please italicize genus names, even in figures, and check for general grammar.
Figures and tables need a more throughout labeling.

Reviewer 3 ·

Basic reporting

Enjoyed reading the manuscript which is represented clearly and the study had found to be well designed. The references are sufficient and represent a wide range of the subject. The statistical analisys are well done followed by great representation with tables and figures. The objective of the study is well explained along the written introduction leading to a defined hypothesis that at the end of the manuscript - data collection, analysis, are found to be relevant to hypothesis.

Experimental design

As wrote before the study design was found to be planned according to the aims and scope of the study .
The methods as well the information are well described - with details, that allow other scientists to follow the study and use in their own studies. The study and the results can be compared with other studies as the units are represent on #/ dry soil basis the standard tool in field studies.

Validity of the findings

The manuscript was readable - clear, fluent, meaningful and will be of a great benefit for scientists in the field.

Additional comments

A nice clear and greatly present manuscript.

Reviewer 4 ·

Basic reporting

The manuscript would greatly benefit from English language editing services to improve clarity.

Experimental design

It appears that “wild fruit forest” refers to two particular species of plant, and it is unclear whether these plants form two separate groupings within the elevation gradient structure (and thus accorded the descriptor of “forest”), or whether they are intermingled.

The methods section must outline how soil parameters (pH, EC, N, P, etc) were measured; a citation alone is insufficient. Currently, this material cannot be replicated.

Line 133: Are the “four basic directions” cardinal directions?

Lines 139-142: This is insufficient. At a minimum, outline the methods used and then include a summary data table for each parameter.

Lines 146-148: This method appears remarkably similar to Jenkins—please cite accordingly.

Validity of the findings

The introduction does not build an argument that justifies a hypothesis that nematode abundance, trophic abundance, and diversity will be highest at middle elevations. Please provide findings from the studies cited to support this choice, or consider widening the hypothesis to suggest that one elevation zone may prove to have the greatest abundance and diversity, as opposed to selecting an option.

If soil chemical parameter data is used to form the basis of analyses presented (in this case, co-occurrence networks), a summary table of soil chemistry data must be included. Otherwise, data interpretation cannot be supported.

The discussion could use considerable contextualization. Carefully consider how nematode indices are used to describe diversity in elevational gradients, and what their interpretive limits are. Since in the faunal diagram most points tend to cluster, does this denote differences or similarities in functional diversity? What other studies have shown similar patterns, and what environmental conditions contributed to these findings? Conversely, find a series of studies that showed diverging patterns and demonstrate how these systems differ. Since sampling occurred at a single time-point, justify how elevation change makes a more compelling argument than, say, periodic introduction of fruit sources? Nematodes constitute only one phylum in the soil foodweb—is it reasonable to state that soils can be characterized and/or described by this one group of organisms, or that elevational gradients simply contribute to a more nuanced understanding of nematode ecology?

Lines 189-190: Please explain why “soil multifunctionality” was selected as opposed to directly using the soil parameters as metadata? Please justify how this type of analysis can be meaningfully compared with other studies that provide raw soil data.

Additional comments

Line 58: The phrase “significantly 1000 times faster” is unclear. Consider applying ranges of values to the parameters you mention.

Lines 208-218: All R packages must have citations.

---

## Round 0.2 · accepted · Accept

Dear Dr. Zhang, I congratulate you on the acceptance of this paper and hope that you will continue to study soil nematode communities in the future. This is a very interesting topic, but it is not worth evaluating only the abiotic effects on nematodes. Much more important are the interactions with fungi, bacteria and plants. I hope that in future articles you will consider these effects.

·

Basic reporting

This study demonstrates that soil nematode communities and their network structures vary significantly with elevation and tree species in the Tianshan wild fruit forest. Environmental factors strongly shape trophic group composition, while nematode network complexity is closely linked to soil multifunctionality. These findings highlight the importance of nematode-based monitoring for long-term assessment of soil health and ecosystem sustainability in this unique forest system.

Experimental design

The methodology of this study is comprehensive and well-structured. By integrating nematode abundance, diversity, ecological indices, and co-occurrence networks across six elevation levels, it captures both structural and functional aspects of the soil food web. The inclusion of two key tree species and soil chemical parameters adds ecological depth, making the findings highly relevant for understanding and monitoring soil health in mountain ecosystems

Validity of the findings

The findings of this study are highly reliable, as they are grounded in a rigorous multi-level analysis that links nematode community structure, ecological indices, and network complexity with key environmental factors. The consistent patterns observed across elevations and host tree species lend strong validity to the conclusions and highlight the robustness of the approach.

Reviewer 2 ·

Basic reporting

Paper is now more readable.

Experimental design

Methodology description is now clearer, but the authors may wish to indicate the model of microscope used.

Validity of the findings

No comment

Reviewer 4 ·

Basic reporting

Language is clear. Explanation and contextualization is sufficient and appropriate.

Experimental design

Experimental design is good. Methods, along with supporting documentation, are complete.

Validity of the findings

Discussion and contextualization is clear and consistent with study findings.